# Relationship between Carbon Sequestration and Soil Physicochemical Parameters in Northern Campeche, Mexico

**Carlos A. Chan-Keb** [1], **Claudia M. Agraz-Hernández** [2,*], **Román A. Pérez-Balan** [1], **Oscar O. Mas-Qui** [1], **Juan Osti-Sáenz** [2] and **Jordán E. Reyes-Castellanos** [2]

[1] Facultad de Ciencias Químico-Biológicas, Universidad Autónoma de Campeche, San Francisco de Campeche 24085, Mexico; carachan@uacam.mx (C.A.C.-K.); roaperez@uacam.mx (R.A.P.-B.); al049387@uacam.mx (O.O.M.-Q.)

[2] Instituto EPOMEX, Universidad Autónoma de Campeche, San Francisco de Campeche 24029, Mexico; jostisaenz@gmail.com (J.O.-S.); joereyes@uacam.mx (J.E.R.-C.)

*  Correspondence: clmagraz@uacam.mx

**Abstract:** For decades, mangroves have been exposed to various pressures, resulting in the loss of large swathes around the world. For this reason, ecological restoration actions are presented as alternatives to recover mangroves that can restore their ecosystem services while helping to mitigate climate change's effects. Mangroves are crucial, as they capture and sequester carbon in biomass and soil, highlighting their importance in environmental conservation and in the fight against climate change. In this research, the amount of carbon sequestered in a mangrove area restored eight years ago and its relationship with soil physicochemical parameters were evaluated and compared to those of a reference forest. Soil cores were collected at a depth of 30 cm from both sites, and in situ measurements of physical chemistry were made at different depths. In addition, soil salinity, bulk density, and carbon concentration were determined. The results revealed a similar amount of carbon sequestered both in the reference forest (BR) ($470.17 \pm 67.14$ Mg C/ha) and in the restoration area (RA) ($444.53 \pm 86.11$ Mg C/ha) ($p > 0.05$). A direct relationship was observed between carbon sequestration and soil depth. In the case of the RA, a direct relationship was found between carbon sequestration and soil salinity. In conclusion, the results of this study indicate that the behavior of carbon sequestration in soil is determined by the physicochemical conditions in both the BR and the RA, as well as by the presence or absence of vegetation.

**Keywords:** restoration; mangroves; ecosystem services; blue carbon; salinity

## 1. Introduction

Mangroves form a coastal ecosystem that exhibits high productivity; they provide a variety of ecosystem services, such as carbon sequestration [1]. Therefore, they are considered to play an important role in mitigating the global warming caused by greenhouse gases (GHGs) owing to their ability to store carbon. This is due to the high productivity of plants and the low decomposition of organic matter in flooded soils and under reduced conditions [2,3].

However, in recent years, mangroves have faced enormous pressures, such as land use change for unsustainable agricultural, livestock, and tourism practices, poorly developed aquaculture, road construction, and high rates of deforestation worldwide [4–6]. In addition, they have suffered the impacts of hurricanes, storms, and droughts associated with climate change [7], mainly due to greenhouse gas emissions generated by various human activities. It is relevant to mention that the disturbance caused by deforestation in mangroves not only implies the loss of aboveground biomass, but also, after the disturbance, significant emissions of carbon dioxide ($CO_2$) and methane ($CH_4$) into the atmosphere [4].

In this regard, Donato et al. [3] determined in 2011 that the destruction of mangroves caused by human activities could release between 73 and 440 million tons of $CO_2$ annu-

ally, contributing to climate change by increasing greenhouse gas emissions. According to the Intergovernmental Panel on Climate Change (IPCC) in 2006 [8], atmospheric concentrations of $CO_2$, $CH_4$, and $N_2O$ have increased significantly over the past 200 years by 30%, 145%, and 15%, respectively. Therefore, the conservation and restoration of mangroves are of paramount importance, as these ecosystems store large amounts of carbon (~1000 Mg C/ha), outperforming boreal forests (~350 Mg C/ha), temperate forests (~349 Mg C/ha), and tropical forests (~230 Mg C/ha) [3,9].

On the other hand, the carbon sequestration capacity of mangroves exhibits great variability at the global, regional, and local levels [10–13], since the atmospheric $CO_2$ fixation capacity in mangroves depends mainly on the density and productivity of the forest, the basal area, height and age of the trees, the topography, the hydroperiod, the physicochemical characteristics of the interstitial water and the soil, as well as the sediment dynamics and photosynthetic efficiency of each species [14,15].

These previously described conditions define the high vulnerability of mangroves to land use change caused by anthropogenic activities, which generate negative effects on the carbon cycle due to the modification of the litter decomposition rate, sedimentation, loss of cover, and plant biomass. These lead to an increase in methane emissions and induce more reducing conditions, resulting from a decrease in the redox potential due to oxygen depletion in the soil. This effect becomes more pronounced when discharges of wastewater from urban, agricultural, livestock, and industrial sources are introduced into wetlands [16,17].

In Mexico, studies related to carbon sequestration in wetlands, specifically in mangroves, have been carried out by Masera et al. in 2001 [18] and Ordóñez-Díaz in 2008 [19], who have estimated a carbon sequestration of 282 Mg C/year for the country's protected wetlands. Of this amount, 177 Mg C/year corresponds to the capture in the biomass (live plant biomass), and 105 Mg C/year corresponds to the carbon stored in the soil (soil and dead trees, fallen branches).

Specifically for the State of Campeche, a carbon storage of 2316 Mg C/ha was determined in the mangroves of Estero Pargo, Bahamita, Ciudad del Carmen, and in the Laguna de Términos Fauna and Flora Protection Area [20]. This storage of organic carbon is indirectly related to the depth of the soil.

Paleoenvironmental and palynological studies carried out in the Los Petenes Biosphere Reserve, located in the State of Campeche, have evidenced changes in vegetation because of sea level rise, droughts during the late Holocene, decreases in sedimentation rate, and marked deforestation events. These changes are a consequence of timber extraction and maize cultivation during colonial times, as well as the presence of the Mayan culture [21].

In recent years, the degradation of mangroves in the state of Campeche has been caused by changes in the hydrological pattern; hurricanes, the construction of communication roads; timber exploitation; and, in the case of Los Petenes Biosphere Reserve, Campeche, the illegal extraction of hunting, logging, and construction of rustic roads [22]. Given the situation mentioned above, the objective of this research was to establish the relationship between the amount of carbon sequestered and soil physicochemical parameters through a soil profile 30 cm deep in an area of mangrove ecological restoration that was carried out 8 years ago. This was conducted to determine the soil physicochemical variables that influence carbon sequestration in a restored mangrove area.

## 2. Materials and Methods

### 2.1. Study Area

The current study was conducted in the northern region of the State of Campeche, where the Los Petenes Biosphere Reserve (RBP) is situated (Figure 1), which is part of an ecoregion that includes the Ría Celestún Biosphere Reserve and the El Palmar State Natural Protected Area, in the state of Yucatán. These areas contribute to the high diversity of flora and fauna, as well as to the uniqueness of some of their ecosystems, in particular the Petenes mangroves and seagrasses. This is considered the state with the best conservation

rate in the entire Mexican country. Mangroves occupy approximately 50% of the terrestrial surface area within the Los Petenes Biosphere Reserve. Four species have been identified, with *Rhizophora mangle* predominantly located along the coastline, displaying a distinct fringe forest physiognomy. The internal forests are dominated by *Avicennia germinans*, exhibiting two distinct physiognomic types—one leaning toward a basin forest type and the other characterized as a dwarf forest type. Both forest types demonstrate the presence of *Laguncularia racemosa* and *Conocarpus erectus*.

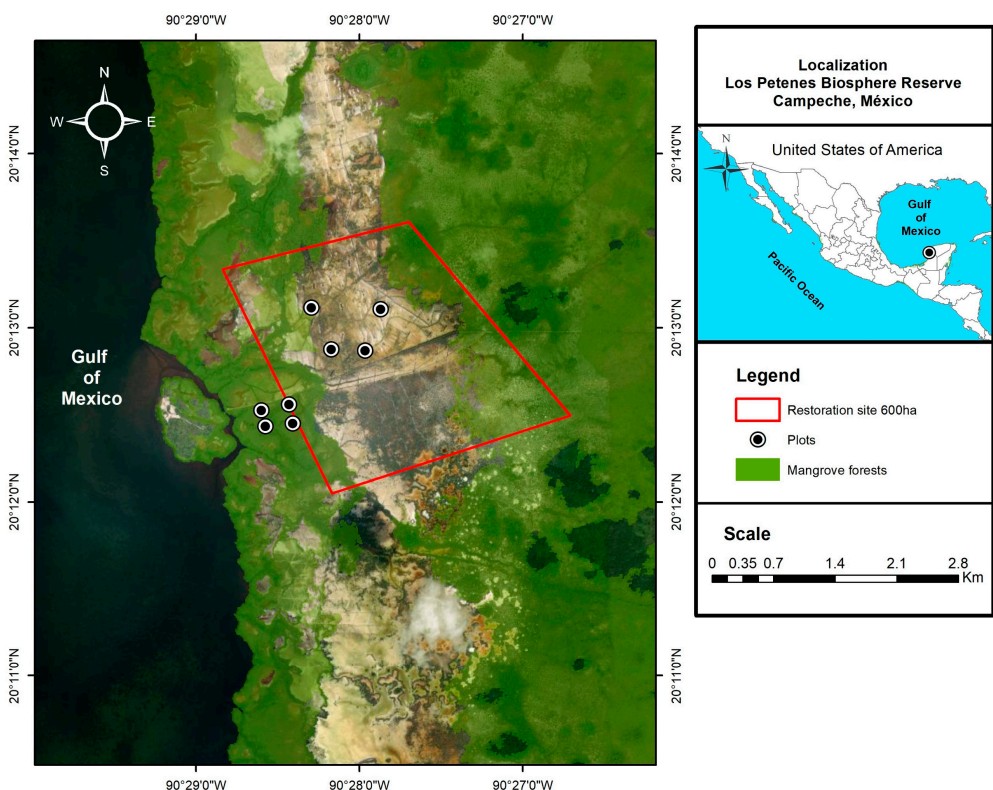

**Figure 1.** Geographical location of the Los Petenes Biosphere Reserve in Campeche and the sampling sites.

In this study, a 600-hectare area of deceased mangrove was selected for the application of ecological restoration techniques (20°12′48.79″ N, 90°28′3.06″ O) and methods, in March 2012. The process included hydrological rehabilitation achieved through the construction of an artificial canal network, actively involving four Mayan communities. Additionally, reforestation actions were implemented using the aerial seeding method with *Avicennia germinans* propagules. The soil texture in the restoration area was characterized as loam, with 21–36% sand, 38–63% silt, and 13–26% clay. The reference forest was categorized as silt-loam, with 11–27% sand, 64–65% silt, and 9–23% clay.

In April 2019, during the course of this investigation, the survival rate of the reforested propagules reached 90%, with densities of 4000 individuals per hectare and average heights of 2.5 ± 0.5 m.

Adjacent to the restored area, a preserved mangrove forest was chosen as a reference. This forest displayed a fringe physiognomy with 2150 trees/ha, 16 m²/ha of basal area, and a height of 8.9 m, featuring the presence of *A. germinans*, *R. mangle*, and *L. racemosa* (20°12′29.95″ N; 90°28′32.54″ O).

Carbon stores were quantified, and soil parameters were measured in 2019 in a restoration area (RA) and a reference forest (BR). The reference forest was adjacent to the restoration area (Figure 1). Four plots in each forest were selected to adequately represent each site and carry out the sampling. Soil samples were extracted at a depth of 30 cm using nucleators constructed with PVC pipes of 15.24 cm diameter, following the criteria

described in reference [23]. Measurements were taken at 5, 10, 15, 20, 25, and 30 cm depths per core, and a total of 24 samples per site were collected.

## 2.2. Determination of pH, Redox Potential, Temperature, and Salinity

In situ determinations of pH, redox potential, and temperature in the soil cores, every 5 cm in depth, were carried out using an IQ150 multiparameter (Loveland, CO, USA). Subsequently, the cores were cut at different depths to determine salinity and bulk density in the laboratory. Salinity was determined using an aqueous solution from the soil and the use of an A&O refractometer (ATAGO, Inc., Bellevue, WA, USA), with a measurement range of 0 to 100 practical salinity units (PSUs) [24].

## 2.3. Determination of Bulk Density

Soil bulk density was determined using the cylinder method of known volume $(g/cm^3)$ [25]. This physical parameter was evaluated for each core at depths of 5, 10, 15, 20, 25, and 30 cm in the soil. For the collection of samples at different depths, a cylinder with a diameter of 15.2 cm was used, which was filled with soil. After the samples were collected, they were subjected to 105 °C in a convection oven until a constant dry weight was reached. The bulk density was calculated by dividing the weight of the dry soil by the volume of the cylinder, as described in the following formula (Equation (1)):

$$Dap = \frac{Ps}{Vs} \text{ in g/cm}^3 \tag{1}$$

where *Dap* = bulk density of the soil in $g/cm^3$, *Ps* = dry weight of soil in g, and *Vs* = occupied volume of soil mined in situ in $cm^3$.

## 2.4. Determination of Soil Carbon

The organic carbon analysis was performed via the dry combustion method, based on Dumas' principle as described by Bremner [26], using a FLASH 2000 elemental analyzer (Thermo Fisher Scientific Inc., Waltham, MA, USA). In each sample generated for each core and depth, 10 mg of soil in silver capsules was weighed using a Mettler Toledo XP6 microbalance (Mettler Toledo S.A. de C.V, City México, México) with an accuracy of 0.001 mg. Subsequently, the samples were digested with HCl in a 1:1 ratio to remove the inorganic carbon. The samples were dried on a heating plate at 50 °C, sealed, and then fed into the elemental analyzer. To obtain accurate results, the elemental analyzer was calibrated using standards (sulfanilamide and methionine) and a target.

## 2.5. Quantification of Organic Carbon Sequestration in Soil

The total carbon sequestered content in the soil (Mg C/ha) was determined for each depth interval (5, 10, 15, 20, 25, 30 cm) using Equation (2), and then summed up to obtain the total carbon sequestered content at a depth of 30 cm

$$\text{Mg C/ha} = \text{Bulk density (g/cm}^3) \times C\ (\%) \times \text{depth (cm)} \tag{2}$$

where Mg C/ha = megagrams of carbon per hectare; C (%) = carbon percentage.

## 2.6. Statistical Analysis

A database was generated with the physicochemical parameters of the soil (bulk density, temperature, pH, redox potential, salinity, and carbon sequestration) for each site (ecological restoration area and reference forest) and for each depth interval (5, 10, 15, 20, 25, and 30 cm). A two-way analysis of variance was applied to determine the variation between sites and as a function of soil depth in soil physicochemical parameters. In this analysis, the normality of the physicochemical variables was validated using the method of Shapiro and Wilks (1965), and the homoscedasticity had a significance level of $\alpha = 0.05$. Since the normal distribution assumption was not met, the data were transformed using the

Box–Cox method [27]. Post hoc analysis was realized with the least significant difference (LSD) Fisher's test.

Finally, simple and multiple linear regression analyses were carried out between soil physicochemical parameters (bulk density, temperature, pH, redox potential, salinity) as independent variables and carbon sequestration as a dependent variable. Statistical analyses were performed with a significance level of $\alpha = 0.05$, using Minitab 19 software (Minitab, Release 19.2020.1, LLC, in the United States). Sigma Plot 11.0 (SigmaPlot®, Release 11, San Jose, CA, USA) was used to create the graphs.

## 3. Results

### 3.1. Organic Carbon Sequestration

The carbon sequestration in the restoration area (RA) and the reference forest (BR) did not show significant differences ($p > 0.05$) between the study sites (Figure 2, Table 1), despite the recording of a higher concentration in the reference forest (BR). This higher concentration in the reference forest compared to the restored forest may have been associated with the high soil carbon rates that characterize old-growth forests [3] and by their stronger forest attributes. In addition, the dominance of *A. germinans* and presence of *R. mangle* and *L. racemosa* create suitable conditions for carbon sequestration [28–30]. The soil carbon sequestration at a depth of 30 cm in the mangroves in this study was similar to that reported at different sites in the state of Campeche by authors such as [1,31,32] (Table 1). This could have been due to environmental conditions, such as hydroperiod behavior, soil dynamics, and mangrove species productivity [33–35]. On the other hand, the carbon sequestration results were higher than those reported in some mangrove forest sites in Mexico [12,23,36,37], the United States, Costa Rica [38], Indonesia, Bangladesh, and Mozambique [14,39].

In contrast to the results of [23] in mangroves of the Mexican Pacific, low carbon sequestration due to anthropogenic pressure and loss of forest attributes over time was reported.

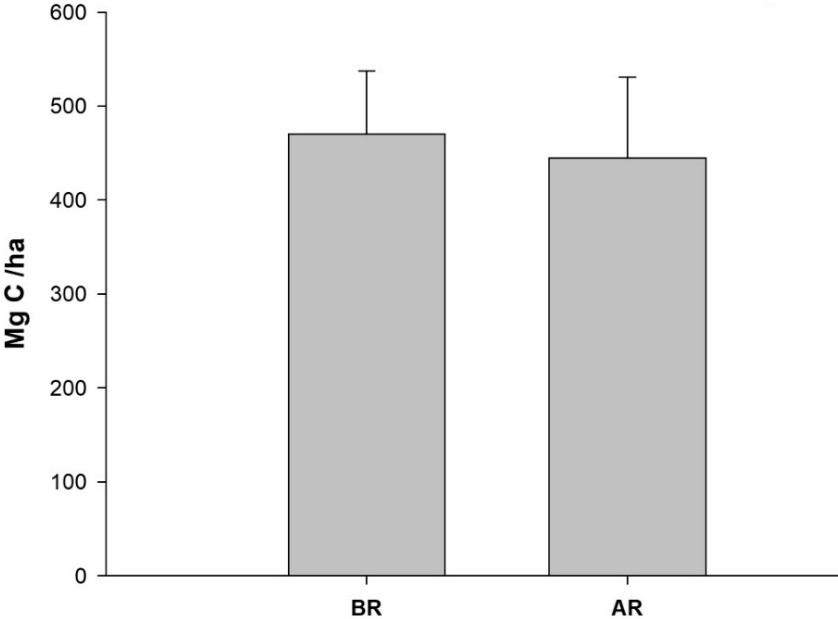

**Figure 2.** Average amount of carbon sequestration in MgC/ha (mean ± standard deviation) at two study sites located in the Petenes Biosphere Reserve, northern Campeche, Mexico. BR: reference forest, and RA: restoration area.

**Table 1.** Carbon sequestration at different depths in mangrove forest soils around the world.

| Place | Depth (cm) | Organic Carbon (Mg C/ha) | Reference |
|---|---|---|---|
| Ohio, USA | 0–24 | 90.3 | [38] |
| Costa Rica | 0–24 | 68.2 | [38] |
| Indonesia | 0–10 | 28 | [39] |
| Mozambique | 0–30 | 53.3 | [14] |
| Bangladesh | 0–30 | 50.8 | [39] |
| Mexican Caribbean | 0–30 | 83.5 | [12] |
| Nayarit, Mexico | 0–20 | 89.7 | [23] |
| Nayarit, Mexico | 0–20 | 36.2 | [37] |
| Veracruz, Mexico | 0–20 | 969 | [36] |
| Campeche, Mexico | 0–50 | 332 | [31] |
| Campeche, Mexico | 0–10 | 174.6 | [32] |
| This study Reference Forest (BR) | 0–30 | 470.17 | |
| Restoration Area (AR) | | 440.53 | |

### 3.2. General Behavior of Soil Physicochemical Parameters

The apparent densities observed at the two study sites were $0.21 \pm 0.01$ g/cm$^3$ for BR and $0.56 \pm 0.06$ g/cm$^3$ for RA (Figure 3b). According to [40], the bulk densities of organic soils in wetlands typically fluctuate in the range of 0.2 to 0.3 g/cm$^3$, while mineral soils have bulk densities of 1 to 2 g/cm$^3$. Therefore, the results obtained in this study correspond to organic soils. However, RA showed a trend from organic to inorganic soil. The variations in bulk density at the study sites were due to differences in the organic and inorganic composition of the soils, their geomorphology, and the presence of minerals such as CaCO$_3$, which are typical of the soils of the Yucatan Peninsula [24]. For this reason, the study sites had low bulk densities due to the high content of organic matter, as is the case with BR. This similar behavior is described in the work of [41] in wetland soils in Veracruz, Mexico, and in the study by [32] in mangrove soils of the Petenes Biosphere Reserve in Campeche, Mexico.

The study area exhibited salinity concentrations (Figure 3c) typical of areas where the hydrological system is characterized by the availability of fresh water, as well as surface and interstitial water during the rainy season [42].

The general behavior of the temperature recorded was homogeneous (Figure 3d) and similar to that reported in tropical mangrove ecosystems [43–45].

The pH recorded at the two sampling sites (AR and BR) in this study was acidic (Figure 3e). This may have been e due to the formation of hydrogen sulfide through the contribution of seawater and the oxide-reduction condition of the soil, as well as the formation of humic and fulvic acids during the decomposition of organic matter. Similar behaviors have been reported by [46–48].

The highest values of the redox potential were recorded at the AR (Figure 3f) due to the hydrological rehabilitation carried out in 2012, which allowed a more controlled hydrological pattern. On the other hand, the potential values recorded in this study are like those reported in the state of Campeche [32,49] and in the Laguna de Términos Flora and Fauna Protection Area [50].

### 3.3. Relationship between Carbon Sequestration and Soil Physicochemical Parameters

Overall, carbon sequestration showed a direct correlation with depth, i.e., carbon (%) and carbon sequestration showed trends of greater availability with increasing depth (Figure 3a, Table 2). The trend described above may have been due to the high concentration of salinity and the redox potential that limits microbial development [51]. Similar behavior has been reported for mangroves in Campeche [31,47], Ca Mau [52], Mekong Delta, Vietnam [53], North Vietnam [54], and Indonesia [55].

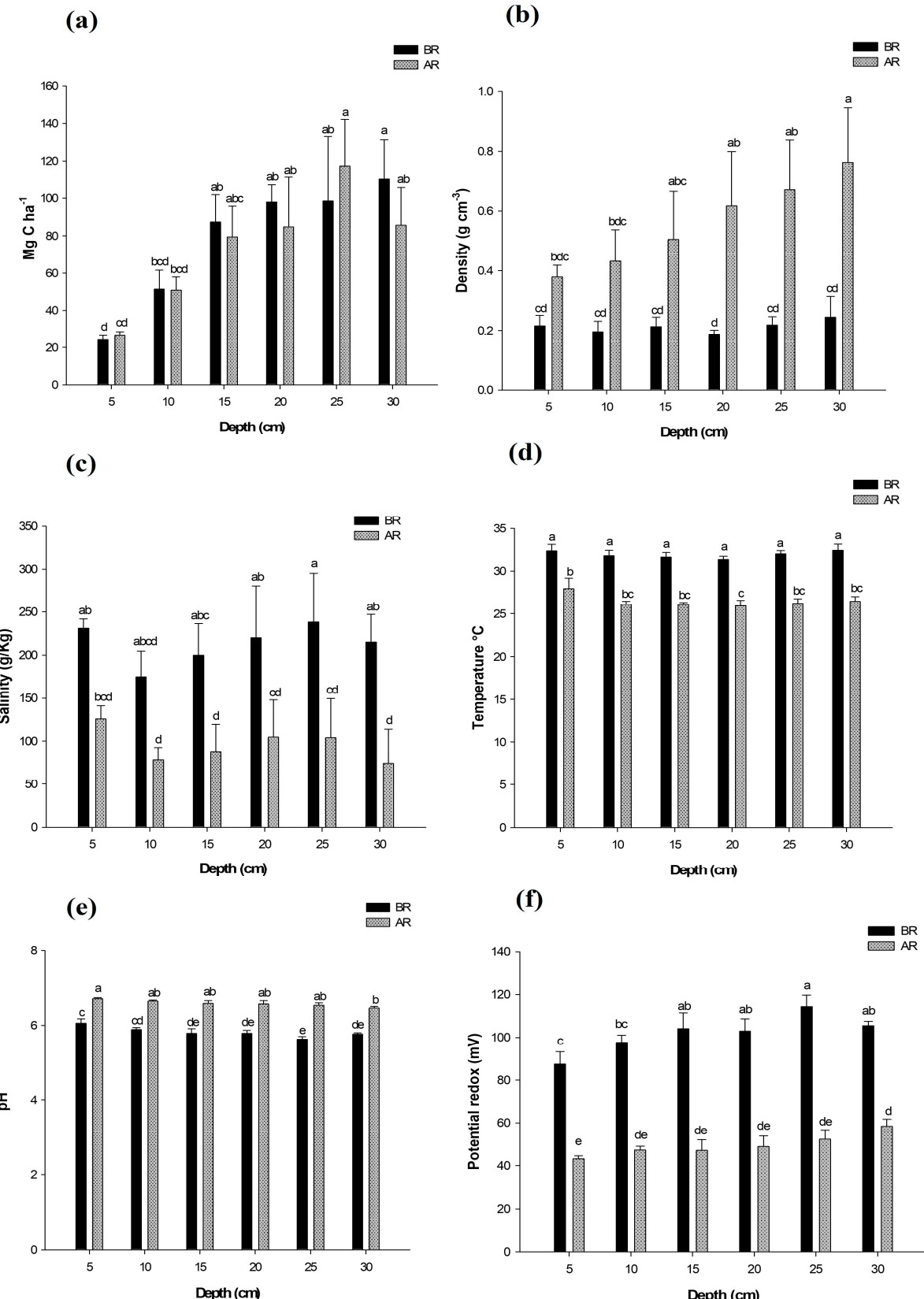

**Figure 3.** Carbon sequestration (**a**), density (**b**), salinity (**c**), temperature (**d**), pH (**e**), redox potential (**f**) (average ± standard deviation) at different depths in sites of the reference forest and restoration area. Error bars that do not share a letter are significantly different ($p < 0.05$).

**Table 2.** Simple linear regression analysis of carbon sequestration and soil physicochemical parameters of the reference forest and mangrove restoration area located in the north of the state of Campeche, Mexico.

| Site/Variable Dependent (Y) | Equation | $R^2$ | $p$ |
|---|---|---|---|
| Reference Forest | | | |
| SC (Mg C/ha) | $Y = 20.39 + 3.313 \, X_1$ | 0.41 | 0.001 |
| SC (Mg C/ha) | $Y = 571.3 - 85.20 \, X_2$ | 0.23 | 0.024 |
| CO (%) | $Y = 33.97 - 49.51 \, X_3$ | 0.21 | 0.031 |
| Restoration Area | | | |
| SC (Mg C/ha) | $Y = 34.93 + 0.4175 \, X_4$ | 0.48 | 0.0001 |
| SC (Mg C/ha) | $Y = -11.9 + 2.441 \, X_1 + 0.4690 \, X_4$ | 0.79 | 0.0001 |
| CO (%) | $Y = 28.50 - 28.49 \, X_3$ | 0.71 | 0.001 |
| PS (cm) | $Y = 9.161 + 14.14 \, X_3$ | 0.23 | 0.018 |

SC: carbon sequestration, CO: organic carbon, PS: soil depth, $X_1$: soil depth in cm, $X_2$: soil pH, $X_{3}$: bulk density in g/cm$^3$, $X_4$: soil salinity in g/kg, with a significance level $\alpha = 0.05$.

In the reference forest, an inverse relationship was shown between carbon sequestration and soil pH (Table 2); this was due to the high production of organic matter from leaf litter, which favors the formation of organic acids (humic and fulvic) as part of the natural degradation of leaves in the process of the biogeochemical carbon cycle. Likewise, the influence of the tides on the mangrove forest favors the formation of hydrogen sulfides, which is also reflected in a decrease in pH [49].

The bulk densities observed at the sampling sites show a significant inverse correlation with the percentage of soil CO. This relationship was validated using a simple linear regression analysis (Table 2).

In the specific case of the restoration area, carbon sequestration showed a significant direct correlation between salinity and soil depth ($p < 0.05$) (Table 2). This could be due to hydrological changes at the time of disturbance of the area, a condition that favors increased water evaporation and salinity concentration. In degraded forests, carbon sequestration losses of 90% and 66% are common [56–58]. Despite this, in the RA, hypersalinization led to carbon sequestration by influencing the mineralization of compounds and organic matter, preventing their degradation and sequestering organic matter in the soil.

## 4. Discussion

Mangrove forests provide a wide variety of ecosystem services with very high productivity. The restoration of mangroves affected by anthropogenic activities could be the type of intervention needed to recover the productivity of their ecosystem services. In this study, a high carbon sequestration capacity was demonstrated by the restoration area (RA), despite having undergone restoration eight years ago. This capacity can be attributed to the accumulation of organic matter that gave rise to the mangrove forest prior to the fragmentation of the ecosystem, which was caused by the construction of a rustic road for accessing an archaeological site named Jaina. As a result, hypersalinization, neutral tending to acidic pH, and reduced soil conditions occurred, all of which influenced the mineralization of organic components and the inhibition of microbial activity [59,60]. These phenomena limited the decomposition of organic matter and the release of carbon in the form of methane, which favored carbon sequestration at the site [22].

The highest carbon sequestration values in this study coincide with those reported by [61], since this author indicated that the mangroves of the Gulf of Mexico, where the study area is located, exhibit the largest carbon stocks in the country due to their high primary productivity and degree of conservation. Thus, a large amount of organic carbon is stored in the soil [36,62].

Another factor that can influence carbon sequestration is the chemical composition of the organic matter of mangrove species stored in the soil. In 2015, Kauffman et al. [62] determined, for the mangroves of the Pantanos de Centla, larger organic carbon stores at

a depth >100 cm, with more than 1000 Mg C/ha, where the soil stored up to 80% of the carbon stocks of the entire ecosystem.

The inverse correlation between bulk density and carbon percentage could be due to the high amount of organic matter that accumulates in the mangrove forest that comes from primary production. These results are similar to studies carried out by [37] in mangroves in Nayarit, Mexico, and those by [36] in coastal wetlands of Veracruz, where they showed an inverse relationship between %CO and bulk density. Thus, this study validates their findings.

Regarding salinity levels, because the Petenes Biosphere Reserve (where the study was carried out) is in the north of Campeche State, and the largest contribution of water is of a direct marine type, it is also characterized as having only availability of fresh water. Surface and interstitial water pathways are those used by the water in the rainy season [42]. However, it should be noted that salinity measurements were carried out in soil and not in interstitial water, unlike other studies. Therefore, another factor that influences the concentration of salinity in the soil of coastal regions and wetlands is the retention and precipitation of salts exerted by the soil. Thus, the accumulation of salts is associated with soils with textures showing a high clay content, low permeability, and reduced leaching, promoting the concentration of salts and ions such as $SO_4^{2-}$, $Cl^-$, $Na^+$, $Ca^{2+}$, and $Mg^{2+}$, which are predominantly salt-forming agents [63]. Salinity in coastal wetlands is basically determined by hydroperiod, tidal fluctuations, seawater salinity, changes in sea level, freshwater inputs from rivers, precipitation, and groundwater [64–66].

It is important to emphasize the importance of temperature measurement, since mangroves are very susceptible to temperature changes, affecting their productivity and, therefore, carbon sequestration [45,47].

For pH, Ref. [67] mentions that oxide-reduction conditions are related to hydrology, that is, water residence time, frequency of flooding due to tidal effect, and urban and/or industrial wastewater inputs. In this way, the trend of sediment reduction in the RBP is conditioned by the contributions of seawater (tides), the annual hydrometeorological cycles (precipitation), and the type of anthropic pressure that is exerted in the ecosystem. Because of this, the oxic conditions in the redox potential at the sampling sites in the RBP are a consequence of the low residence time of the water caused by the ebb and flow of the tide, as well as the low inflow of fresh water during rainfall, and by interstitial and groundwater.

The data shown in Table 2 on carbon sequestration may differ from those of studies that associated low salinity values in interstitial water and greater carbon sequestration in the soil [31,37,41]. However, this occurs in healthy mangrove soil surfaces, as they favor the development of certain forest attributes (base area, density, and height) and the increased production of leaf litter, which is why this study determined that, in degraded mangrove soils, high concentrations of salinity prevent the release of carbon in the form of $CH_4$ and $CO_2$ and therefore favor carbon sequestration through the process of mineralization of organic matter. In general terms, the importance of the ecological restoration of the mangrove area (hydrological rehabilitation and reforestation) improved the physicochemical conditions of the soil for carbon sequestration and the recovery of the ecosystem services of the mangrove area.

## 5. Conclusions

In general, the highest amount of carbon sequestration was recorded in the reference forest compared to the restoration area; however, there were no significant differences ($p > 0.05$). At different soil depths, no significant differences ($p > 0.05$) were observed in the %CO in the reference forest and the restoration area. However, with respect to carbon sequestration, both sites show a trend of increasing accumulation at greater depths, showing significant differences ($p < 0.05$) between the depth intervals.

It is concluded in this research that the behavior of carbon sequestration in soil is defined by the physicochemical conditions of the soil in the BR and AR, as well as the presence or absence of vegetation. A high capacity for carbon sequestration was demonstrated

by the restoration area (RA). This underscores the significance of ecological restoration in degraded mangrove areas, enabling the restoration of the capacity for carbon sequestration as an ecosystem service.

**Author Contributions:** Conceptualization, C.A.C.-K. and C.M.A.-H.; methodology, O.O.M.-Q.; validation, C.A.C.-K. and C.M.A.-H.; formal analysis, C.A.C.-K.; investigation, J.O.-S. and J.E.R.-C.; writing—original draft preparation, O.O.M.-Q.; review and editing, R.A.P.-B.; supervision, C.M.A.-H. All authors have read and agreed to the published version of the manuscript.

**Funding:** This research received no external funding.

**Data Availability Statement:** The data are available on request.

**Conflicts of Interest:** The authors declare no conflicts of interest.

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
