# Peer review of "Relationship between Carbon Sequestration and Soil Physicochemical Parameters in Northern Campeche, Mexico"

_land, doi:10.3390/land13020139_

Round 1
Reviewer 1 Report
Comments and Suggestions for Authors
The paper investigates the amount of carbon stored in the soil of two mangrove forests (restored and natural) in relation to the physical and chemical characteristics of the soil. Studying the different aspects of carbon sequestration in mangroves is becoming a global priority, especially with escalated degradation of mangroves due to human impacts. The present study is considered as baseline research, by which it provides an account for the levels of carbon stored on the sediments along with their physical characteristics.
The study uses typical methods and procedures, including estimating carbon using an element analyzer and measuring physical and chemical parameters in the field and in the laboratory. The statistical approach is reasonable, and the study follows needed assumptions. The results are presented logically in figures and tables.
However, the study may benefit further from the following suggestions:
1. Discussion: I assume there is a need to include implications of the study on restoration efforts. This is because restoration is mentioned in the introduction and conclusion, but it was not discussed.
2. Methodology: more details about the characteristics of mangroves in the two types will provide clearer picture about these sites. For instance, what are the species of mangrove in these two forests.
3. Line 16: 8 years = Eight years
4. Line 73: (dead soil and trees, …) = soil and dead trees
5. Figure 1: It will be good to include the figure in English instead of Spanish to ensure the consistency with the manuscript language. Further, there is a need to distinguish between the restored forest and the reference one on the map.
6. Line 115: 4 plots = Four plots (typically sentence starts with a word).
7. Line 191: Table I = Table 1
8. Line 217= Figure 3 legend should be below the figure.
9. Lines 257, 262, 269, 271: correct the numbers of the tables (1 or 2).
Reviewer 2 Report
Comments and Suggestions for Authors
A few questions and remarks on the article by Chan-Keb et al. “Relationship between carbon sequestration and soil physico-chemical parameters in northern Campeche, Mexico”.
Mangroves are formed by various species of woody plants. The species of wood plant affects how much litter is added to the soil and at what rate, which affects the quantity of organic carbon in the soil. The woody plant species that formed the mangrove forests are not mentioned in the manuscript. Please provide this information. Include a brief biological feature of these tree species as well.
None of the researched mangrove forests' characteristics are mentioned. What are the mean diameters and heights of tree trunks in the mangrove regions under investigation? What is the average age of the trees? Please provide these data.
Specify when you conducted your study. Month (months) and year.
Provide data on soil texture.
The article does not include Figure 2.
Move the paragraph (containing Table 1) between lines 185 and 200 to the Discussion section. You should compare and discuss your findings with those of other researchers in the Discussion section.
Round 2
Reviewer 2 Report
Comments and Suggestions for Authors
All the issues I cared about were addressed by the authors. The manuscript has been carefully revised.